# Using tree-based ensemble methods to produce a population-based mortality risk score in Ontario, Canada

Steven Habbous[1,2]*, Peter C. Austin[3,4], Shabnam Balamchi[1], Davood Astaraky[1], Roozbeh Yousefi[1], Munaza Chaudhry[1], Erik Hellsten[1]

1 Ontario Health, Toronto, Ontario, Canada, 2 Department of Epidemiology and Biostatistics, Western University, London, Ontario, Canada, 3 ICES, Toronto, Canada, 4 Institute of Health Policy, Management and Evaluation, University of Toronto, Toronto, Ontario, Canada

* Steven.habbous@ontariohealth.ca

## Abstract

### Introduction

Risk adjustment is critical in observational epidemiology to control for confounding of the exposure-outcome relationship. Accurate prediction of outcomes, such as mortality, can improve risk adjustment. In the present study, we compared logistic regression with a range of tree-based ensemble methods to predict 1-year mortality in the general population of Ontario, Canada.

### Methods

Ontario adults (age 18 years and older) who were alive as of January 1, 2022 were included. Using a window of up to 3 years, various measures of health and health-care utilization were captured from administrative databases. To predict 1-year mortality, we applied logistic regression, random forests, extremely randomized trees, adaptive boosting, gradient boosting, extreme gradient boosting, Newton boosting, and CatBoost. All models also included age and sex. Performance was evaluated using the area under the ROC curve (AUROC), the area under the precision-recall curve (PR-AUC), the Brier score, and a quantile-based version of the Integrated Calibration Index (ICI), reported in the 30% test set. Feature importance was assessed using CatBoost's internal model structure, supplemented with permutation feature importance, explainable boosted machines, and marginal effects.

### Results

A total of 12,080,801 Ontarians were included and 121,951 (1.0%) died within 1 year. Logistic regression showed excellent discrimination (AUROC 0.926; PR-AUC 0.256) and acceptable calibration (ICI 0.0022). The best model was CatBoost, which had the best discrimination (AUROC 0.933, PR-AUC 0.280) and calibration (ICI 0.0003).

**Data availability statement:** Ontario Health is prohibited from making the data used in this research publicly accessible if it includes potentially identifiable personal health information and/or personal information as defined in Ontario law, specifically the Personal Health Information Protection Act (PHIPA) and the Freedom of Information and Protection of Privacy Act (FIPPA). Due to these legal and ethical restrictions, data will not be made publicly available. However, upon request (Datarequest@ontariohealth.ca), data de-identified to a level suitable for public release may be provided.".

**Funding:** The author(s) received no specific funding for this work.

**Competing interests:** The authors have declared that no competing interests exist.

In sensitivity analyses of the CatBoost model, including more detailed definitions of cancer (to include its subtype) and chronic kidney disease (defined using serum creatinine instead of diagnostic codes) produced modest improvements in PR-AUC (0.290), along with substantially improved calibration amongst the highest-risk (70–100%) individuals. The most influential model-building feature was age. Residence in long-term care and receipt of palliative care was associated with the largest marginal effects.

## Conclusion

The machine learning model CatBoost yielded the most accurate predictive model for 1-year mortality using individual comorbidities and additional measures of healthcare utilization for the general population. These findings demonstrate that machine learning methods can enhance risk adjustment efforts in observational studies, leading to more accurate confounder control and better support for health policy and epidemiologic research.

## Introduction

Risk prediction models are common in the medical industry, often guiding treatment decisions. Examples include risk equations that predict the probability of kidney failure, the probability of lung cancer given a pulmonary nodule, the probability of post-operative pulmonary complications, and the 10-year risk of cardiovascular disease [1–4]. Traditionally, such risk scores have been constructed using standard statistical regression techniques like linear regression, logistic regression, or Cox proportional hazards regression.

Machine learning methods may be better suited than traditional statistical methods when the number of potential predictors is large, when there is potential for multi-collinearity, when synergistic or antagonistic effects (e.g., interactions) between predictors exist, and when associations are non-linear. Traditional statistical models would require the onerous and subjective task of manually coding interaction terms. Tree-based methods automatically consider interactions and non-linearity by virtue of subsequently splitting the data into partitions as the depth of the trees increases, and like logistic regression, can also output a predicted probability of the outcome.

Despite this, applications of machine learning methods in public health are limited [5,6]. In the present study, we constructed models for estimating the risk of all-cause 1-year mortality in the general adult population in Ontario, Canada. The output from these models can be used as a single variable for risk adjustment or confounder control, or as a variable of interest to understand the burden of illness in a population. We use a range of tree-based ensemble methods, uncovering some of the black-box features of machine learning methods and providing additional exposure for epidemiologists and health system analysts interested in predictive modeling [7].

## Methods

### Cohort creation

This retrospective population-based study was conducted in Ontario, Canada. The cohort included all Ontario residents (with valid Ontario postal code and valid unique identifier) who were alive as of January 1, 2022. Persons with sex coded as M or F and aged 18–105 as of the index date (January 1, 2022) were included (N = 12,080,801). Reporting follows STROBE for observational studies (STROBE Checklist in S1 File).

### Outcome

The outcome of interest was 1-year all-cause mortality from the index date (January 1, 2022). The date of death was captured from the Registered Persons Database.

### Predictors of 1-year mortality

Two sets of predictors were considered. The first set of predictors included age (continuous), sex, and comorbidities (chronic conditions) captured from a hospital setting adapted from the Charlson Comorbidity Index using the updated codes for HIV/AIDS and renal disease as outlined by Glasheen et al [8,9]. The second set of predictors expanded on the first by including additional measures of healthcare utilization that could be captured outside of the hospital setting.

### Comorbidities – Chronic conditions

Comorbidities were categorized using ICD-10 codes from hospital visits within three years prior to the index date [8,10]. We used the Discharge Abstract Database (DAD) for inpatient visits, (allowing up to 25 diagnostic codes per visit), and the National Ambulatory Care Reporting System (NACRS) for ambulatory hospital visits (allowing up to 10 diagnostic codes per visit) (Appendix 1 in S2 File for codes). Comorbidities included myocardial infarction, congestive heart failure, peripheral vascular disease, cerebrovascular disease, dementia, chronic pulmonary disease, connective tissue/rheumatoid disease, peptic ulcer disease, mild liver disease, diabetes without complications, diabetes with complications, hemiplegia/paraplegia, moderate-to-severe liver disease, mild-to-moderate renal disease, severe renal disease, primary cancer, metastatic cancer, HIV, and AIDS (HIV with opportunistic infection).

### Measures of healthcare utilization

We used the Ontario Health Insurance Plan (OHIP) database to capture health and healthcare utilization based on physician billing codes that could plausibly be related (positively or negatively) to mortality. Flags were created using a lookback of 1 year and included billing codes related to obesity, long-term care/chronic care residence, mental health, house calls, American Society of Anesthesiologists (ASA) physical status classification (V, IV, III, none), chronic pain, fibromyalgia, palliative care, smoking cessation, homecare, substance use/abuse, critical care, hyperbaric treatment, vaginal delivery or delivery by Cesarean section, major burns, and amputation (Appendix 2 in S2 File). Acute hospital-based indicators included hip fracture, delirium, and hospital-acquired pressure injury. Indicators of healthcare utilization included the number of unique visits with the healthcare system (OHIP), the number of admissions (DAD), and the number of outpatient hospital encounters (NACRS). Activity of Daily Living (ADL) score was obtained from the Continuing Care Reporting System (long-term care; complex continuing care) and InterRAI-Home Care databases. The ADL is a measure of an individuals' abilities to perform self-care tasks (ranges from 0–28). The most recent score available during the preceding year was used, with missing values assigned a "missing" category.

### Data pre-processing

Numeric predictors (age, number of healthcare utilization visits, ADL score) were retained as continuous predictors. Persons missing an ADL score (e.g., those not in a setting to complete an InterRAI survey) were assigned a separate

category. These variables were not rescaled, as the machine learning models employed were based on decision trees, which are invariant to monotonic transformations and operate effectively on the rank order of predictor values. Categorical variables were transformed using one-hot encoding (dummy-coded to 0/1) to ensure compatibility with the machine learning algorithms.

## Models

**Base model: Logistic regression.** The baseline set of models was logistic regression, which is estimated using maximum likelihood estimation (MLE). MLE identifies the parameters that maximize the log-likelihood function. In machine learning implementations of logistic regression, the objective is typically framed as minimizing the negative average log-likelihood function.

**Tree-based ensemble methods I: Bagging.** The first set of ensemble methods involved bagging (bootstrap aggregation) methods [11]. These methods randomly split the data across $n$ base estimators (e.g., decision tree classifiers), which can be trained independently and therefore in parallel. In standard bagging, bootstrap samples (sampling with replacement) are drawn from the training data, and each base estimator uses all available features.

In random forests, instead of using all available features to split each node within a tree, a random subset of features is selected at each split, increasing diversity and reducing overfitting.

Extremely randomized trees (ExtraTrees) are similar to random forest, but instead of picking the best variable to split on among the random sample of features at each node, ExtraTrees chooses the splitting variable by picking the best variable to split according to random split thresholds produced for each feature.

All these bagging-based methods aim to reduce variance and improve generalization by introducing diversity among the base learners, which are ensembles of strong learners (e.g., deep trees).

**Tree-based ensemble methods II: Boosting.** The second set of ensemble methods involved boosting methods, which sequentially construct shallow trees (e.g., weak learners), with each iteration aimed at correcting the errors of the previous model [11].

In adaptive boosting (AdaBoost), observations misclassified by previous learners are assigned higher weights, so that subsequent learners focus more on the difficult-to-classify cases. Each weak learner is associated with its own weight, and the final prediction is a weighted combination of all learners.

In gradient boosting, instead of reweighting misclassified observations, learners are fit to the gradients of a loss function, which quantify the degree of error. Here, the loss function is based on residual errors (observed minus predicted from previous weak learner), and weak *regressors* are fit to the residuals, which are real-valued rather than binary. Gradient boosting uses first-order (single derivative) information.

Newton boosting extends gradient boosting by incorporating second-order derivatives (the Hessian) to adjust weights, enabling faster convergence. Extreme gradient boosting (XGBoost) is a type of Newton boosting, designed for greater computational efficiency and allowing for regularized loss functions.

Finally, we used CatBoost, a newer Newton boosting method optimized for datasets with many categorical variables, especially those with high cardinality (many levels for a categorical variable, such as primary cancer type). CatBoost uses oblivious decision trees instead of standard decision trees (the same splitting criterion is used for all splits occurring at the same level) (https://catboost.ai/) [12–14]. Although CatBoost handles categorical variables and missing values natively, in our application, all categorical variables were non-missing and already one-hot encoded due to their dichotomous nature.

## Assessing model performance

The dataset was randomly split into 70% training and 30% test, stratified by the outcome to ensure equal proportions of the outcome in both datasets. Model performance was assessed in the test set while considering performance in the training set to identify evidence of overfitting (e.g., marked degradation in performance between training and test datasets).

**Discrimination.** The primary metric was the area under the receiver operating characteristic (AUROC) curve. We also considered the area under the precision-recall curve (PR-AUC). Whether AUROC or PR-AUC is more appropriate when the outcome is unbalanced (e.g., the outcome of mortality is uncommon) remains a matter of discussion, so we provide both, favouring the AUROC [15,16]. We supplemented this with the Brier score (categorical equivalent of mean squared error, computed as the average squared distance between the predicted probability and the observed dichotomous outcome as 0/1) [17]. For all models, we inspected the distribution and the range of the predicted probabilities (ideally ranging from 0.01 to 0.99), how close the mean predicted mortality was to the observed mean mortality (ideally they are the same), and whether the AUROC, PR-AUC, and Brier scores differed between the training and test datasets (ideally they are the same).

**Calibration.** While the Brier score also incorporates calibration [17], we also assessed calibration by visual inspection of the calibration curve, plotting centiles of the predicted probability of mortality versus the observed probability of mortality [18,19]. Large deviations of the calibration curve from the diagonal line of perfect calibration, if driven by sparsity (e.g., few observations had a predicted probability in that range), may cause one to over-interpret deviations from calibration [20]. Given that the `lowess()` function was computationally expensive, we calculated a metric comparable to the Integrated Calibration Index ($ICI_{eq}$) by computing $\frac{\sum n_i \left( \left| \overline{Y}_{pred} - \overline{Y}_{observed} \right| \right)}{N}$, interpreted as the weighted average of the absolute difference between mean observed and expected outcomes, where $n_i$ is the size of the bin and $N$ is the total sample size (100 bins were selected) (Appendix 3 in S2 File for Python code). We also supplemented this by comparing mean(y) with mean($\hat{y}$) (the closer the better; calibration-in-the-large) and Cox's calibration slope and intercept [21]. To calculate Cox's calibration slope and intercept, we used a logistic regression of the observed outcome regressed on the logit-transformed predicted probability of the outcome [21–23]. A well calibrated model is expected to have a slope of 1 (a unitless quantity) and an intercept of 0. Lastly, for visual support we used Kaplan-Meier plots stratified by predicted probabilities of the outcome, expecting the observed event rates to match the predicted event rates in each group [24].

### Sensitivity analyses

**Definition of select comorbidities.** We refined the definition of renal disease and primary cancer to leverage our more detailed data assets (Appendix 4 in S2 File for details). For renal disease, we used serum creatinine from the Ontario Laboratory Information System (OLIS) to calculate eGFR using the 2021 CKD-EPI formula that is not race-adjusted [25]. This was done because ICD-10 codes have demonstrably poor sensitivity for capturing chronic kidney disease (CKD), resulting in underestimation of this highly prevalent condition [26]. Persons with no serum creatinine were assumed to have no kidney disease. For primary cancer, we used the Ontario Cancer Registry (gold standard for cancer diagnoses), categorizing each diagnosis using the SEER recode classification system. The SEER codes were unchanged, with the following exceptions due to small sample sizes: 22050 (pleura) and 22060 (trachea, mediastinum, other respiratory organs) were combined with 22030 (lung and bronchus); 35041 (other acute leukemia), 35023 (other myeloid/monocytic leukemia) 35031 (acute monocytic leukemia), and 35041 (other acute leukemia) were combined as "Other leukemia"; and 33011 (nodal) and 33012 (extranodal) were combined into "Hodgkin lymphoma".

**Source of comorbidity.** Comorbidities were primarily dichotomized as present/absent. In sensitivity analysis, we further distinguish by hospital visit type as: 1) none (comorbidity absent); 2) ambulatory visit only (NACRS); 3) inpatient visit only (DAD); and 4) ambulatory and inpatient visit (both DAD and NACRS).

**Data splitting.** In a sensitivity analysis, we compared the performance of the best-performing model when the same model was constructed using 10-fold cross-validation.

**Hyperparameter tuning.** We started with arbitrary values and fine-tuned manually for select models (e.g., number of trees, depth of trees, learning rate). Due to resource constraints, we did not perform a randomized or grid search of hyperparameters.

 

**Validation cohort.** We identically constructed a second validation cohort comprised of Ontario adults alive as of January 1, 2024. We applied the best-performing model to this cohort to assess generalizability and potential drift.

## Explainability

**Feature importance: Internal to the model.** Methods using gradients (gradient boosting, extreme gradient boosting, Newton boosting, CatBoost) do not use entropy or Gini impurity to determine a split. Instead, leaf values are updated using gradients information and indicate to what extent the models' predictions should be increased or decreased. Using CatBoost's internal structure, a feature's importance was calculated as the sum of the absolute changes in prediction (leaf value changes) caused by splits on that feature, weighted by the number of samples passing through those splits, and then normalized so the importances sum to 100 across all features (formula: https://catboost.ai/docs/en/concepts/fstr#regular-feature-importance).

**Feature importance: External to the model.** Model-agnostic methods do not require the inner workings of the model, only its output. We provide two approaches: permutation feature importance that is more common to the machine learning literature, and marginal effects that is more common among the statistical literature.

Permutation feature importance (PFI): for each column, the values are randomly shuffled within the dataset and the difference between the original metric and the permuted one is estimated [27]. We have chosen the AUROC as the metric for estimating the PFI.

Marginal effects: To estimate the overall effect of each feature on 1-year mortality, we computed the average marginal effect (AME) and the relative marginal effect (RME) using recycled predictions [28]. For each categorical variable we computed the predicted probability of the outcome ($\hat{y}$) under two scenarios: 1) when all individuals in the cohort had a value $= 0$ ($y_0 = \hat{y}_{x_i=0}$); and 2) when all individuals in the cohort had a value $= 1$ ($y_1 = \hat{y}_{x_i=1}$), where $x_i$ is the categorical variable. For a numeric variable (age, number of healthcare visits), $\hat{y}_{x_i=0}$ is the value of the variable as-is ($y_0 = \hat{y}_{x_i=x_i}$) and $\hat{y}_{x_i=1}$ is the value of the variable $+ 1$ ($y_1 = \hat{y}_{x_i=x_i+1}$). The AME was computed as a mean of the differences $\left( AME = \frac{\sum y_1 - y_0}{N} \right)$, interpreted as the mean absolute change in the outcome due to a 1-unit increase in the predictor. The RME was calculated a mean of the relative difference $\left( RME = \frac{\sum \frac{(y_1 - y_0)}{y_0}}{N} \right)$, interpreted as the relative average change in the predicted probability associated with a 1-unit increase in the predictor. The 95% confidence interval for AME and RME was computed using the standard error of the difference times the critical t-statistic where alpha $= 0.05$.

**Explainable boosting machines.** Explainable boosting machines (EBM) are tree-based ensemble methods that sacrifice accuracy for explainability [11,29]. EBMs leverage generalized additive models (GAMs), whereby each predictor in the model is a function learned from the data rather than a single parameter. Each function is smoothed and can be non-linear. These functions are then added together to produce the linear predictor, and like other regression models, are linked with the outcome through some link function (e.g., logit). With EBMs, each feature's function is modeled using a shallow tree using gradient boosting and a small learning rate. Pairwise interactions can be added to GAMs manually, but the EBM approach automates this after first fitting the main effects, and only retains interactions if they improve performance.

## Software

Cohort creation was performed using SAS v9.4 (SAS Institute Inc., Cary, NC) and all analyses were performed in Python using the `statsmodels.api-v0.14.2` for logistic regression with MLE, `scikit-learn-v1.5.2` for random forest, ExtraTrees, (extreme) gradient boosting, `catboost-v1.2.7` for CatBoost, and `interpret.glassbox-v0.6.10` for EBMs in Jupyter Notebook (v4.0.11). `numpy-1.26.4` was used because version <2 is required for CatBoost and older

versions are incompatible with EBM. Feature details are presented in Appendix 5 in S2 File. Sample code is provided in Appendix 6 in S2 File (training and testing a CatBoost model).

### Privacy and ethics

Research ethics approval was not required as per the Ontario Health privacy assessment as this work was performed for the purpose of quality improvement and no identifying information was obtained. This study was compliant with section 45(1) of PHIPA (Ontario Health is a prescribed entity); thus, patient consent was not required.

## Results

A total of 12,080,801 Ontarians who were alive as of January 1, 2022 were included. Half (n = 6,194,717; 51.3%) were female, the mean age was 49.0 (SD 18.6) years, and 121,951 (1.0%) died within 1 year (Table 1).

The most prevalent comorbidity was diabetes without complications (n = 513,176; 4.3%), followed by primary cancer (n = 257,092; 2.1%) and diabetes with complications (n = 236,554; 2.0%) (Table 1). OLIS identified nearly 6 times more persons living with CKD (n = 345,477; 2.9%) compared with CIHI. The OCR identified 1.39% of the population having had a cancer diagnosis in the last 3 years compared with the standard CCI definition (2.13%). Among other health encounters, flu vaccination was the most prevalent (12.8%), followed by breast cancer screening (8.8%) and a mental health flag (3.7%).

The risk of 1-year mortality varied by comorbidity type or health encounter type, with the highest mortality rates associated with receipt of palliative care (36%), residence in LTC (27%) dementia (26%), pressure injury (25%), delirium (21%), metastatic cancer (20%), stage 4–5 renal disease (16–17%), and congestive heart failure (16%) (Table 1). The probability of death for primary cancer was 9.3% but varied from 1% (e.g., thyroid or testicular cancer) to >30% (e.g., mesothelioma, pancreatic cancer, brain cancer, esophageal cancer) (Appendix 7 in S2 File).

### Model performance

Logistic regression (MLE; model 1A) produced AUROC 0.926, PR-AUC 0.256, Brier Score 0.0085, and $ICI_{eq}$ 0.0022, performing similarly to random forest (model 1B) (Table 2). The best-performing model was CatBoost with a learning rate of 0.05 and a maximum tree depth of 6 (model 1H, Table 2), which had the highest AUROC (0.933), highest PR-AUC (0.281), lowest Brier Score (0.0083), and lowest $ICI_{eq}$ (0.0003) (Fig 1A). Cox's calibration intercept was not significantly different from 0 (slope = 0.017; p = 0.09) and Cox's calibration slope was not significantly different from 1 (slope = 1.005; p = 0.08).

### Sensitivity analysis

Using Model 1H as the comparator, we assessed a series of sensitivity analyses (Table 2, section 3). Defining cancer from the Ontario Cancer Registry improved performance in PR-AUC only, whether the cancer diagnoses were one-hot encoded (model 2A) or ordered-target encoded (Model 2B-2C). Including the source of comorbidity (e.g., inpatient, outpatient) (Model 2D), defining CKD using OLIS as stages 2–5 (Model 2E), or using 10-fold cross-validation instead of a 70%/30% training/test split (Model 2F) did not appreciably affect performance.

**Survival analysis.** A Kaplan-Meier plot of the predicted probabilities from model 1H in the test dataset on the time until death also revealed separation of the curves that was evident as early as 1-month follow-up and continued through the 12 months (Fig 2B). The model over-estimated the risk of death among the group with a predicted probabilities between 70 and 100%, which contained only 321 persons (Table 3). We therefore hypothesized that people with the highest actual likelihood of dying can be captured by incorporating individual cancer diagnoses from the OCR. Using Model 2E, despite no noticeable improvement in performance, we found better discrimination for the 70–100% risk group (Figs 1B and 2B), although a small degree of miscalibration was observed [Cox's intercept 0.028 (p = 0.006) and Cox's slope 1.009 (p = 0.002)].

**Table 1. Prevalence and crude risk of 1-year all-cause mortality by comorbidity in full cohort (N = 12,080,801).**

| Demographics | N | % | N dead | % dead |
|---|---|---|---|---|
| Age group, years | | | | |
| 18–24 | 1,177,234 | 9.74 | 633 | 0.05 |
| 25–29 | 1,037,589 | 8.59 | 712 | 0.07 |
| 30–34 | 1,098,660 | 9.09 | 970 | 0.09 |
| 35–39 | 1,053,164 | 8.72 | 1,151 | 0.11 |
| 40–44 | 967,615 | 8.01 | 1,399 | 0.14 |
| 45–49 | 943,636 | 7.81 | 1,914 | 0.20 |
| 50–54 | 979,521 | 8.11 | 2,984 | 0.30 |
| 55–59 | 1,066,770 | 8.83 | 5,088 | 0.48 |
| 60–64 | 1,002,826 | 8.30 | 7,691 | 0.77 |
| 65–69 | 847,192 | 7.01 | 9,790 | 1.16 |
| 70–74 | 707,371 | 5.86 | 12,554 | 1.77 |
| 75–79 | 508,074 | 4.21 | 14,742 | 2.90 |
| 80–84 | 338,390 | 2.80 | 17,030 | 5.03 |
| 85–89 | 211,834 | 1.75 | 19,379 | 9.15 |
| 90–94 | 104,698 | 0.87 | 16,572 | 15.8 |
| 95+ | 36,227 | 0.30 | 9,342 | 25.8 |
| Sex | | | | |
| Male | 5,886,084 | 48.7 | 63,073 | 1.07 |
| Female | 6,194,717 | 51.3 | 58,878 | 0.95 |
| **Comorbidity** | **N** | **%** | **N dead** | **% dead** |
| Congestive heart failure | 101,696 | 0.84 | 16,654 | 16.4 |
| Myocardial infarction | 72,944 | 0.60 | 5,073 | 6.95 |
| Cerebrovascular disease | 109,725 | 0.91 | 8,269 | 7.54 |
| Peripheral vascular disease | 44,501 | 0.37 | 4,434 | 9.96 |
| Cardiopulmonary disease | 183,040 | 1.52 | 12,540 | 6.85 |
| Connective tissue/rheumatoid disease | 17,763 | 0.15 | 1,197 | 6.74 |
| Peptic ulcer disease | 46,641 | 0.39 | 2,458 | 5.27 |
| Mild liver disease | 39,257 | 0.32 | 3,614 | 9.21 |
| Moderate-to-severe liver disease | 14,309 | 0.12 | 1,965 | 13.7 |
| Dementia | 49,276 | 0.41 | 12,608 | 25.6 |
| Hemiplegia/paraplegia | 15,004 | 0.12 | 1,506 | 10.0 |
| Diabetes without complications | 513,176 | 4.25 | 23,609 | 4.60 |
| Diabetes with complications | 236,554 | 1.96 | 20,476 | 8.66 |
| Nephritic syndrome | 1,866 | 0.02 | 125 | 6.70 |
| Mild-to-moderate renal disease | 40,958 | 0.34 | 6,899 | 16.8 |
| Severe renal disease (Stage 5, CIHI) | 29,619 | 0.25 | 3,578 | 12.0 |
| Chronic Kidney Disease (OLIS) | | | | |
| Stage 2 (eGFR < 90 mL/min) | 1,215,456 | 10.1 | 29,472 | 2.42 |
| Stage 3a (eGFR 45 to <60 mL/min) | 206,373 | 1.7 | 13,113 | 6.35 |
| Stage 3b (eGFR 30 to <45 mL/min) | 91,025 | 0.8 | 9,500 | 10.4 |
| Stage 4 (eGFR 15 to <30 mL/min) | 32,039 | 0.3 | 5,161 | 16.1 |
| Stage 5 (eGFR < 15 mL/min) | 16,040 | 0.1 | 2,700 | 16.8 |
| Cancer (primary) | 257,092 | 2.13 | 23,937 | 9.31 |
| Cancer (metastatic) | 57,305 | 0.47 | 10,979 | 19.2 |
| Cancer stage (Collaborative Staging) | | | | |
| Missing | 67482 | 0.56 | 6817 | 10.10 |
| 1 | 42987 | 0.36 | 1482 | 3.45 |
| 2 | 24543 | 0.20 | 1361 | 5.55 |

*(Continued)*

| Demographics | N | % | N dead | % dead |
|---|---|---|---|---|
| 3 | 19568 | 0.16 | 2289 | 11.70 |
| 4 | 13796 | 0.11 | 4061 | 29.44 |
| HIV | 1,435 | 0.01 | 62 | 4.32 |
| AIDS | 405 | <0.01 | 30 | 7.41 |

| Other health flag | N | % | N dead | % dead |
|---|---|---|---|---|
| House call | 285,521 | 2.36 | 4,099 | 1.44 |
| Home care | 140,837 | 1.17 | 16,843 | 11.96 |
| Long-term care | 76,119 | 0.63 | 20,363 | 26.75 |
| Palliative | 20,478 | 0.17 | 57,154 | 35.83 |
| Critical care | 354,897 | 2.94 | 25,646 | 7.23 |
| Activities of Daily Living score | | | | |
| 1-5 | 37,986 | 0.31 | 4,848 | 12.8 |
| 6-10 | 40,580 | 0.34 | 6,323 | 15.6 |
| 11-15 | 29,115 | 0.24 | 5,627 | 19.3 |
| 16-20 | 36,998 | 0.31 | 9,396 | 25.4 |
| 21-28 | 38,201 | 0.32 | 12,995 | 34.0 |
| Obesity | 66,984 | 0.55 | 1,095 | 1.63 |
| Mental health | 443,265 | 3.67 | 7,873 | 1.78 |
| Smoking cessation | 106,608 | 0.88 | 1,351 | 1.27 |
| Substance abuse | 68,576 | 0.57 | 1,426 | 2.08 |
| Delirium | 59,555 | 0.49 | 12,449 | 20.90 |
| Hip fracture | 33,865 | 0.28 | 4,719 | 13.93 |
| Fibromyalgia | 34,047 | 0.28 | 476 | 1.40 |
| Amputation | 3,207 | 0.03 | 468 | 14.59 |
| Pressure injury | 3,200 | 0.03 | 809 | 25.28 |
| Vaginal delivery | 242,401 | 2.01 | 81 | 0.03 |
| Cesarean delivery | 113,445 | 0.94 | 31 | 0.03 |
| Flu vaccination | 1,057,086 | 8.75 | 19,312 | 1.83 |
| Breast cancer screening (last 3 years) | 1,543,695 | 12.8 | 7,419 | 0.48 |
| Chronic pain | 1,554 | 0.01 | 46 | 2.96 |
| Hyperbaric | 1,254 | 0.01 | 68 | 5.42 |
| Major burn | 282 | 0.00 | 8 | 2.84 |
| American Society of Anesthesiologists (ASA) physical classification score | | | | |
| III | 391,124 | 3.24 | 6,117 | 1.56 |
| IV | 136,192 | 1.13 | 12,047 | 8.85 |
| V | 5,625 | 0.05 | 679 | 12.07 |

| Unique counts of healthcare use[a] | Mean (SD) | Median (IQR) | 90th percentile | Any (count >0) |
|---|---|---|---|---|
| Any physician billing (OHIP) | 9 (12.6) | 5 (1, 12) | 22 | 9,759,893 (81%) |
| Admissions (DAD) | 0.1 (0.3) | 0 (0, 0) | 0 | 596,620 (4%) |
| Ambulatory visits (NACRS) | 0.7 (4.3) | 0 (0, 1) | 2 | 3,106,316 (26%) |

[a]count of the unique number of service dates [from the Ontario Health Insurance Program (OHIP) database], admission dates from the Discharge Abstract Database (DAD), or registration dates from the National Ambulatory Care Reporting System (NACRS).

CIHI – Canadian Institute for Health Information, includes DAD and NACRS.

IQR – interquartile range.

OLIS – Ontario Laboratory Information System.

SD – standard deviation.

**Table 2. Model performance statistics on the test set.**

| | Model | AUROC (higher better) | PR-AUC (higher better) | Brier score (lower better) | ICIeq (lower better) |
|---|---|---|---|---|---|
| **1** | **Predictors: age + sex + individual comorbidities from Charlson (present/absent) + other measures of healthcare utilization** | | | | |
| A | Logistic regression using maximum likelihood estimation (MLE) [1 min] | 0.926 | 0.256 | 0.0085 | 0.0022 |
| B | Random forest (500 trees, max depth = 10, min samples split = 10, min samples leaf = 10, max features = sqrt) [15 min] | 0.921 | 0.268* | 0.0084 | 0.0021 |
| C | ExtraTrees [52 min] | 0.930 | 0.268* | 0.0084 | 0.0017 |
| D | Gradient boosting (base estimator: max depth 3, min samples/split 10, min samples/leaf 5. Subsequent estimators: n = 100, max depth 3, min samples/leaf 10, min samples/split 10, $\eta = 0.1$) [21 min] | 0.929 | 0.262 | 0.0084 | 0.0014 |
| E | Extreme gradient boosting ($\eta = 0.1$, max depth = 3, subsample = 80%, column sample = 80% [1 min] | 0.930 | 0.270 | 0.0083 | 0.0010 |
| F | Newton boosting ($\eta = 0.1$, max leaf nodes = 10, min samples leaf = 10, no regularization [2 min] | 0.931 | 0.266 | 0.0084 | 0.0008 |
| G | CatBoost ($\eta = 0.1$, min samples leaf = 10, no regularization, 1000 iterations, depth = 10) [22 min] | 0.933 | 0.267* | 0.0084 | 0.0007 |
| H[†] | CatBoost ($\eta = 0.05$, min samples leaf = 10, no regularization, 1000 iterations, depth = 6) [12 min] | 0.933 | 0.281 | 0.0083 | 0.0003 |
| I | CatBoost ($\eta = 0.05$, min samples leaf = 10, no regularization, 1000 iterations, depth = 10) [22 min] | 0.933 | 0.276* | 0.0083 | 0.0004 |
| **2** | **Sensitivity analyses. Base model corresponds to CatBoost model 1H (1000 iterations, depth = 6, $\eta = 0.05$, min samples leaf = 10, no regularization)** | | | | |
| A | Primary cancer diagnosis using OCR (one-hot encoded) [14 min] | 0.932 | 0.287 | 0.0082 | 0.0003 |
| B | Primary cancer diagnosis using OCR (ordered target statistics encoding) [42 min] | 0.933 | 0.287 | 0.0082 | 0.0003 |
| C | Primary cancer diagnosis using OCR (ordered target statistics encoding), early stopping = 100 rounds evaluated on AUROC, splitting the training data into 85% training and 15% validation for early stopping, increased to 3000 iterations [120 min] | 0.934 | 0.285 | 0.0082 | 0.0003 |
| D | Primary cancer diagnosis using OCR (one-hot encoded), categorized comorbidities by source (one-hot encoded) [14 min] | 0.932 | 0.288 | 0.0082 | 0.0003 |
| E[†] | Primary cancer diagnosis using OCR (one-hot encoded), categorized comorbidities by source (one-hot encoded), CKD defined using OLIS (one-hot encoded) [18 min] | 0.932 | 0.290 | 0.0082 | 0.0004 |
| F | 10-fold cross-validation instead of 70−30 train-test split [2.5 hours] | 0.933 ± 0.0011 | 0.279 ± 0.0037 | 0.0083 ± 0.0000 | 0.0005 ± 0.00007 |
| G | Adding cancer stage (I, II, III, and IV) [15 min] | 0.932 | 0.290 | 0.0082 | 0.0003 |
| H | Removed ADL from the model, reducing the number of datasets required [15 min] | 0.931 | 0.274 | 0.0083 | 0.0004 |
| **3** | **Explainable Boosting Machine (EBM) for explainability** | | | | |
| A | EBM (max rounds = 50, $\eta = 0.01$, max leaves = 3, max bins = 255, interactions = 0) [5 min] | 0.926 | 0.230 | 0.0089 | 0.0059 |
| B | EBM (max rounds = 50, $\eta = 0.01$, max leaves = 3, max bins = 255, interactions = 3) [8 min] | 0.927 | 0.239 | 0.0087 | 0.0039 |
| C | EBM (max rounds = 50, $\eta = 0.01$, max leaves = 3, max bins = 255, interactions = 15) [9 min] | 0.926 | 0.249 | 0.0086 | 0.0021 |
| D[†] | EBM (max rounds = 1000, $\eta = 0.01$, max leaves = 3, max bins = 255, interactions = 20) [62 min] | 0.931 | 0.268 | 0.0084 | 0.0008 |

[†]best predictive model in the group. Changing the number of observations in a leaf before a split to 20 or 50 did not change the results.

* some evidence of overfitting (substantially better performance in training set).

AUROC – area under the receiver operator characteristic curve; PR-AUC – area under the precision-recall curve; $ICI_{eq}$ – Integrated Calibration Index equivalent (estimated using centiles); $\eta$ – learning rate; OCR – Ontario Cancer Registry; CKD – chronic kidney disease; OLIS – Ontario Laboratory Information System.

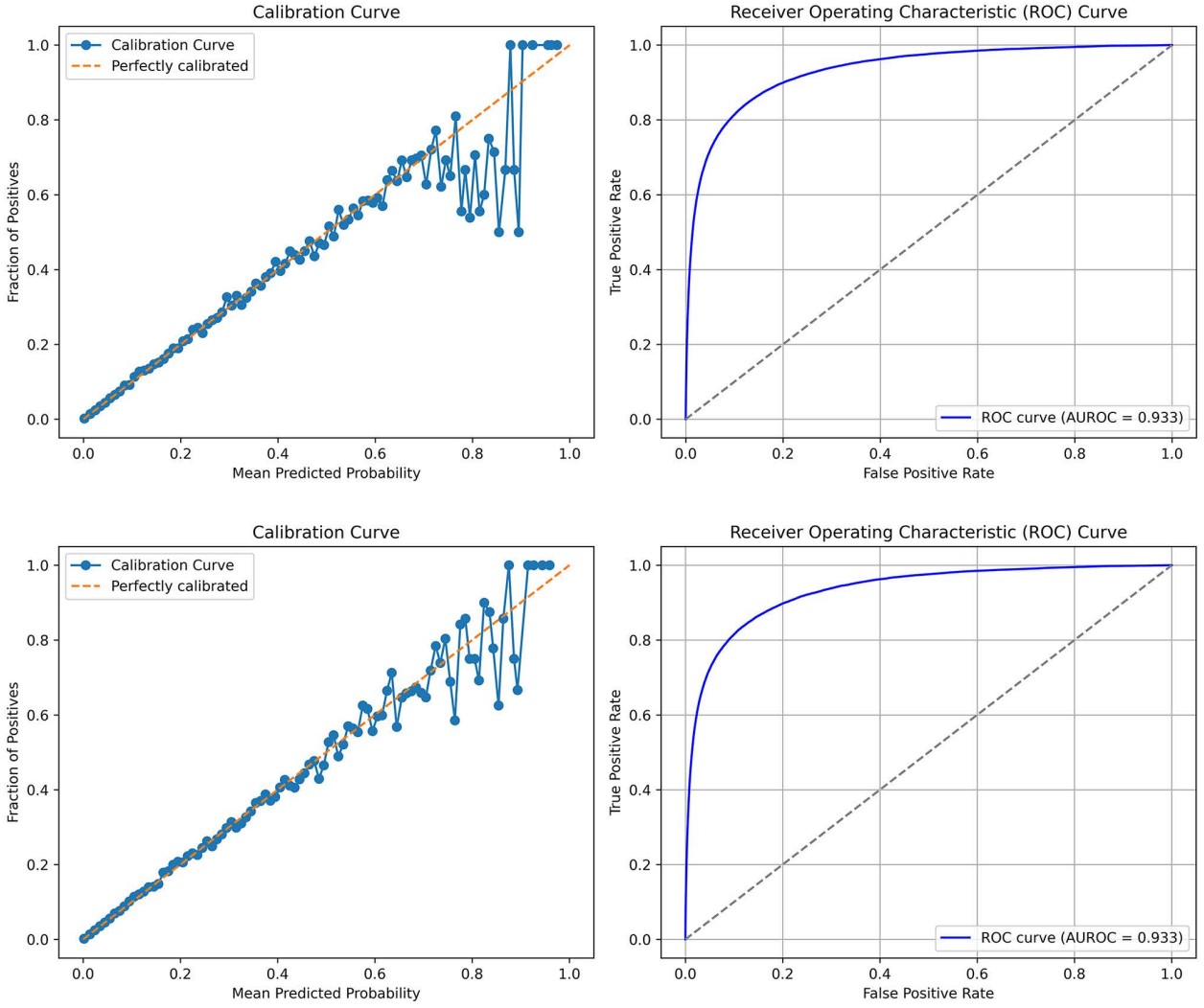

**Fig 1. Calibration curve and Receiver Operator Characteristic (ROC) curves for best performing models. A)** Model 1H (CatBoost; learning rate = 0.05, min samples leaf = 10, no regularization, 1000 iterations, depth = 6); **B)** Model 2E (CatBoost; same as Model 1H, but with the inclusion of primary cancer types, chronic kidney disease stages, and comorbidities categorized by hospital encounter source). AUROC – area under the ROC curve.

## Explainability

Based on feature importance from CatBoost model 1H, age was the most important feature, followed by the number of outpatient hospital visits, sex, the number of encounters with the healthcare system (based on OHIP), ADL score, the number of hospitalizations, palliative care, and breast cancer screening (Fig 3A). Using PFI, a similar set of features was identified (Fig 3B). Using EBM, age was the most important feature and was included in interactions with several other variables (Fig 3C).

To examine the effect of each covariate on mortality risk on an interpretable scale, we report the model parameters from a logistic regression model equivalent to Model 2E in Appendix 8 in S2 File (except comorbidities are dichotomized for simplicity). We also calculated the average and relative marginal effects using model 1H (Appendix 9 in S2 File). Features associated with the highest increased predicted risk of death included receipt of palliative care (AME 4.03%;

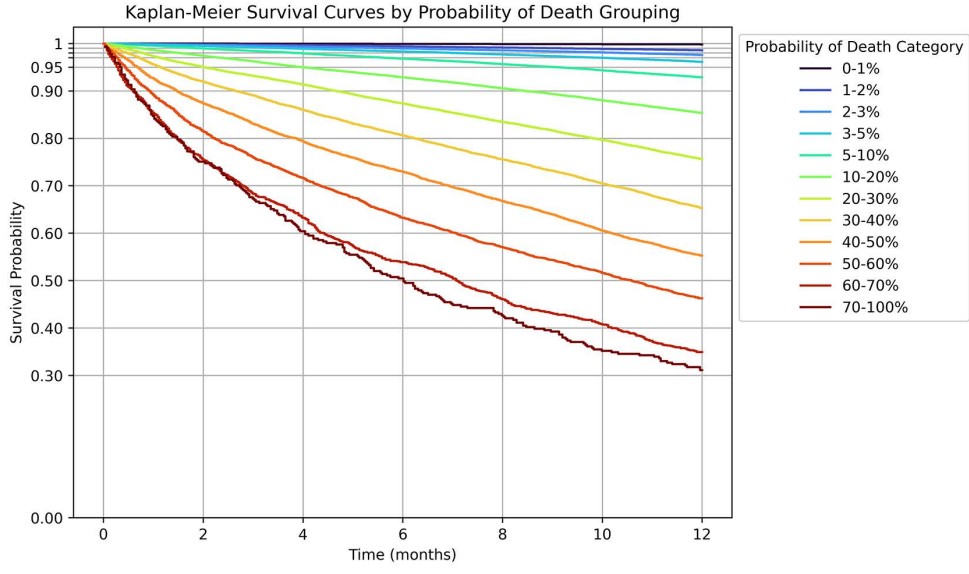

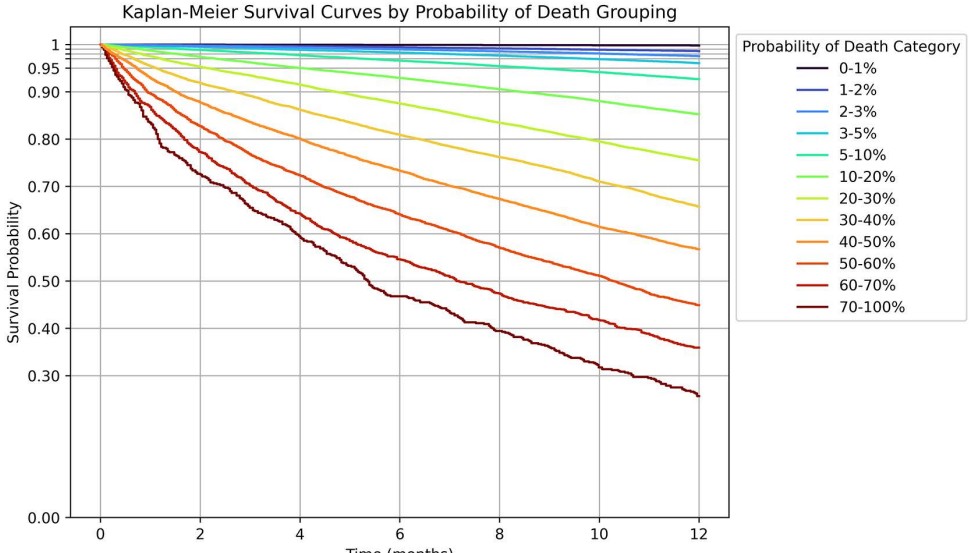

**Fig 2. Kaplan-Meier plots for 1-year all-cause mortality stratified by the predicted probability of death. A)** Model 1H (CatBoost; learning rate = 0.05, min samples leaf = 10, no regularization, 1000 iterations, depth = 6); **B)** Model 2E (CatBoost; same as Model 1H, but with the inclusion of primary cancer types, chronic kidney disease stages, and comorbidities categorized by hospital encounter source).

RME 438%) followed by moderate-to-severe liver disease (AME 2.60%; RME 261%) and metastatic cancer (AME 1.53%; RME 159%). Several features were associated with a lower risk of mortality prediction (e.g., breast cancer screening, ASA score 3, AIDS, influenza vaccination), but the AME was small in magnitude (<0.3% absolute reduction).

## Validation

Applying the best model (2H) to the cohort of Ontarians alive as of January 1, 2024 produced the same AUROC (0.933) and similar Brier Score (0.0077) despite a slightly lower PR-AUC (0.254) and worse calibration ($ICI_{eq}$ 0.0008) due to underestimation of risk (Fig 4).

**Table 3. 1-year mortality by summary score in test set (n=3,624,241).**

| Predicted probability of mortality | Model 1H | | | | Model 2E | | | |
|---|---|---|---|---|---|---|---|---|
| | N | % | N dead | % dead | N | % | N dead | % dead |
| 0-1% | 3,148,906 | 86.88% | 5,803 | 0.18% | 3,158,168 | 87.14% | 5,887 | 0.19% |
| 1-2% | 195,854 | 5.40% | 2,774 | 1.42% | 193,964 | 5.35% | 2,753 | 1.42% |
| 2-3% | 75,427 | 2.08% | 1,807 | 2.40% | 76,087 | 2.10% | 1,816 | 2.39% |
| 3-5% | 67,397 | 1.86% | 2,601 | 3.86% | 62,400 | 1.72% | 2,478 | 3.97% |
| 5-10% | 58,073 | 1.60% | 4,136 | 7.12% | 55,745 | 1.54% | 4,045 | 7.26% |
| 10-20% | 38,092 | 1.05% | 5,573 | 14.63% | 37,186 | 1.03% | 5,460 | 14.68% |
| 20-30% | 17,983 | 0.50% | 4,378 | 24.35% | 17,797 | 0.49% | 4,394 | 24.69% |
| 30-40% | 11,388 | 0.31% | 3,948 | 34.67% | 11,167 | 0.31% | 3,835 | 34.34% |
| 40-50% | 6,461 | 0.18% | 2,889 | 44.71% | 6,488 | 0.18% | 2,828 | 43.59% |
| 50-60% | 3,246 | 0.09% | 1,744 | 53.73% | 3,327 | 0.09% | 1,824 | 54.82% |
| 60-70% | 1,093 | 0.03% | 711 | 65.05% | 1,369 | 0.04% | 853 | 62.31% |
| 70-100% | 321 | 0.01% | 221 | 68.85% | 543 | 0.01% | 412 | 75.87% |

## Discussion

In the present study we examined different machine learning methods for estimating the risk of 1-year mortality using a wide range of health and healthcare indicators. The tree-based ensemble method CatBoost produced the most accurate predictive model and exhibited the best calibration statistics.

Having an accurate predictive model is useful because the predicted probability of mortality can provide some degree of confounder control in epidemiologic studies [30]. Logistic regression to predict the risk of 1-year mortality has been used to produce some of the most common summary scores used for risk adjustment, including the Charlson Comorbidity Index [8], the Elixhauser comorbidity index [31–34], and the Johns Hopkins Aggregated Diagnostic Groups (ADGs) [35]. Although these methods have produced AUROC as high as 0.917 in the general population and remain prognostic many years after their development [35,36], the better discrimination found in our models (AUROC 0.933) suggests that there is opportunity for more accurate risk-adjustment in the general population by leveraging at a minimum only one additional data source (physician billing). Moreover, we use multiple metrics and visual cues to identify the best-performing model. For cohorts where cancer is more prevalent, the performance of the model, and particularly its calibration, would be improved if a more detailed accounting of primary cancer was included.

The improved performance may be driven by several factors. First, we include a wider range of predictors, including those that may be associated with improved mortality (e.g., caesarean or vaginal delivery, influenza vaccination, breast cancer screening). Second, we explicitly included age and sex in the model and implicitly included non-linearity and inter-actions. Compared with tree-based methods, logistic regression would require manual coding of higher-order features for non-linearity, as well as interaction terms, which would be onerous, subjective, and prone to overfitting or underfitting even in the presence of backwards selection. Tree-based methods automatically incorporate interactions (the EBM model supported the importance of interactions particularly with age) and non-linearity by virtue of subsequently splitting the data into partitions as the depth of the trees increase. Like logistic regression, tree-based methods can output a predicted probability of the event. For all comparable models in the general population, CatBoost outperformed logistic regression on all metrics.

## Strengths

Machine learning models have been criticized for being a 'black-box', producing parameters that are unobservable or uninterpretable. The coefficients from a logistic regression output can be interpreted as the relative change in the log odds

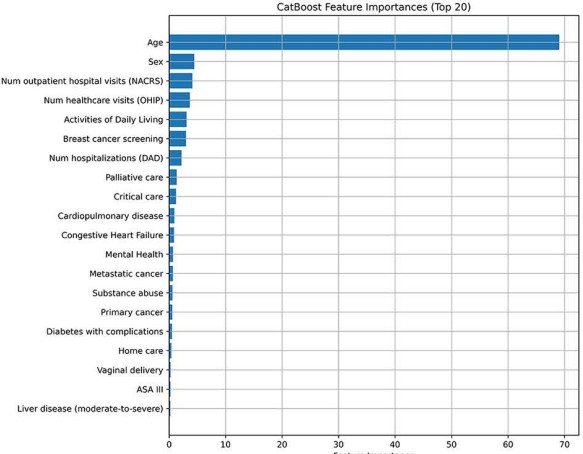

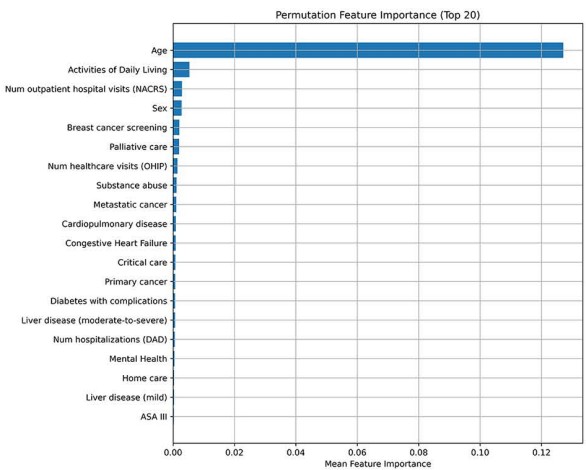

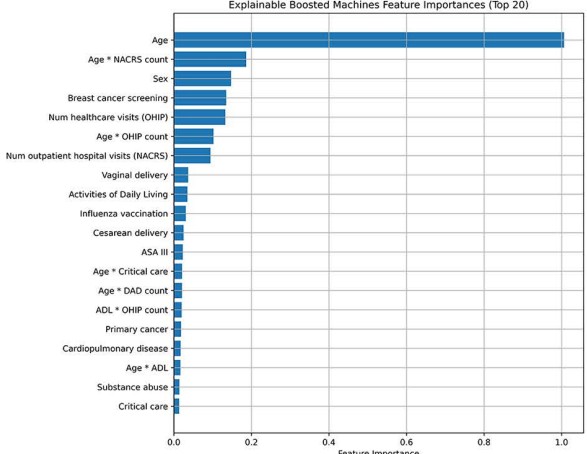

**Fig 3. Feature importance: A) Feature importance from CatBoost (Model 1H), internal from model structure; B) Permutation feature importance (Model 1H), model-agnostic; C) Importance from Explainable Boosting Machine (Model 3D).** Model 1H: CatBoost; learning rate = 0.05, min samples leaf = 10, no regularization, 1000 iterations, depth = 6. Model 3D: explainable boosting machine max rounds = 1000, learning rate = 0.01, max leaves = 3, max bins = 255, interactions = 20. DAD – Discharge Abstract Database (hospitalizations); NACRS – National Ambulatory Care Reporting System (ambulatory hospital visits); OHIP – Ontario Health Insurance Plan (physician billing); ASA – American Society of Anesthesiologists (ASA) physical status classification; ADL – Activities of Daily Living score.

of an event due to the presence of a comorbidity, but the presence of higher-order terms and interaction terms makes interpretability challenging. Moreover, negative coefficients (protective) are sometimes difficult to contextualize and are likely driven by residual confounding. The Elixhauser comorbidity scoring system assigns negative weights to several conditions including obesity and AIDS, features that we also found associated with small negative marginal effects [34]. CIHI's Population Grouper [37] and the Johns Hopkins ADG [36] are both proprietary algorithms, so a deeper understanding of their inner workings are unavailable. A similar issue can also affect machine learning methods, but it is usually hidden by the opaqueness of these methods rather than the modeler's choice. To mitigate this, we have unpacked the inner workings of the model through feature importance rankings (typical for machine learning applications), supplemented with absolute and relative marginal effects (more typical of epidemiology applications). We also provide the code that an

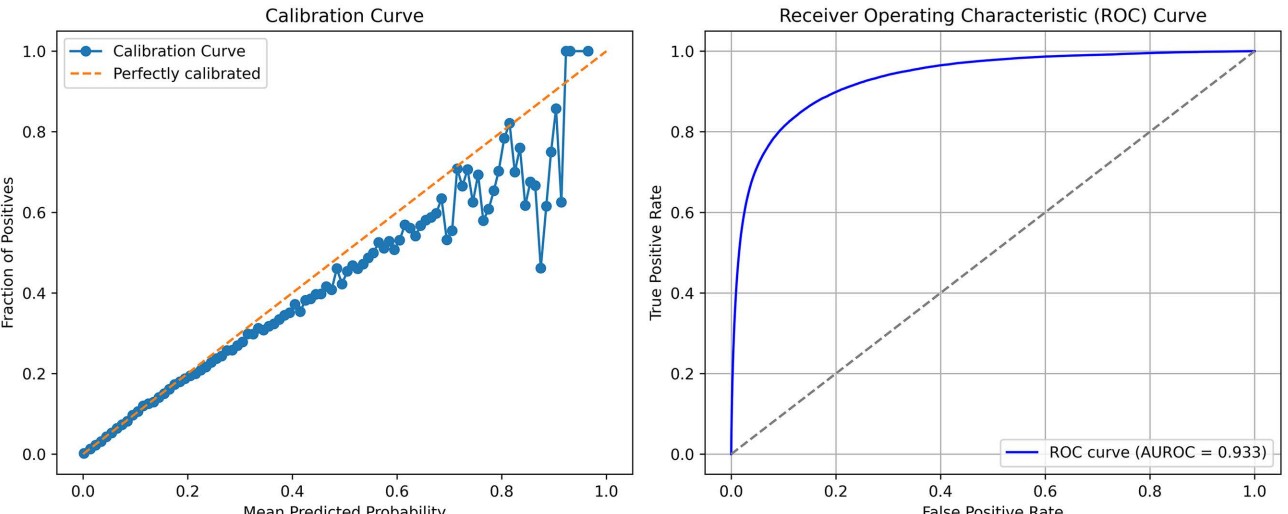

**Fig 4. Calibration curve and Receiver Operator Characteristic (ROC) curves for the best performing models (2E) applied to the 2024 validation cohort.** AUROC – area under the ROC curve.

analyst can use to build their own model using a similar approach, as well as the actual code used to make predictions using the model 'as-is' (File S3 for Model 1H and File S4 for Model 2E).

There is a trade-off between explainability and performance, and many analytics practitioners have sought to meet somewhere in-between (e.g., using 'glass-box' methods optimized for explainability; or reducing the number of features and interactions for parsimony) [29,38,39]. There are a few considerations regarding this trade-off. One is the application: predicting weather or stock prices may demand that accuracy is maximized, while healthcare applications may strive for more explainability (e.g., identifying modifiable risk factors or high-risk groups). Yet another approach would be to take a data-informed rather than a data-driven approach. For example, one study computed a 12-month mortality risk score at the time of admission, and patients with a score exceeding some threshold were referred for palliative care consultation [40]. However, the actual risk score was not provided in the referral because the decision to provide palliative care should be left to the discretion of the clinical staff, not the model. In this example, neither the score itself was important (only the threshold chosen), nor were the individual components that were the main drivers behind risk score. In our study, we promote explainability to lend credibility and support to the model's construction and outputs, and we do so by several means. First, interpretation of calibration plots and supportive visualizations like Kaplan-Meier plots were useful for improving or understanding the model more fulsomely. Increasing model complexity by including primary cancer types, source of comorbidities, and chronic kidney disease stages improved calibration-in-the-large for people with a predicted mortality of 70% or higher. This decision was supported by knowledge (and observation) that different cancer types have vastly different mortality rates, and the intuition that persons hospitalized for a condition are likely at a higher risk of mortality than those who are not [41]. Since this is a small group of people and death is an uncommon outcome, there was little impact on the overall performance of the model based on summary statistics alone (AUROC, PR-AUC, Brier Score, $ICI_{eq}$). Second, we took a top-down approach by identifying factors potentially associated with mortality and then measuring them (this approach facilitates interpretability). This was done by leveraging existing comorbidity scores prevalent in the literature, plus factors known or purported to be associated (positively or negatively) with mortality. Conversely, a bottom-up approach could have yielded better performance at the expense of explainability, whereby the analyst could be removed from the feature engineering process and we would let the model-building process figure it out. Features could

include all diagnostic codes, procedure codes, physician billing codes, laboratory values, and a range of questions from InterRAI surveys beyond ADL, which would yield thousands of features.

Such an approach would be computationally prohibitive. Moreover, there is an unknown practical limit for all statistical measures of discrimination and calibration since 1) mortality is not always predictable; and 2) data are imperfect (e.g., inaccurate diagnoses; migration patterns; incomplete death linkage). However, our models performed very well relative to the theoretical limits ($AUROC_{max} = 1$; $ICI_{eq,max} = 0$) and changes to hyperparameters only produced modest changes in discrimination or calibration. However, there are some limitations that must be acknowledged.

## Limitations

One limitation is the computational resources available. While advancing our ability and capacity to use machine learning methods, our organization is still immature on this front [6]. We adhere to a 'Cloud-First' strategy, but the reliance on a specific-vendor model imposes structural and financial constraints that are ill-suited for the iterative nature of machine learning research. On our Azure Virtual Desktop we have access to eight CPU cores, 137 Gb of RAM, and no GPUs. We were unable to conduct full cross-validation assessments or hyperparameter tuning through grid or randomized search, and it remains possible that a better model fits these data. Despite this, the convergence of different models across all measures of model performance suggest that any further improvements in the model are likely to be small. For the present study this was not a barrier, but for work requiring more sophisticated models, a hybrid approach of cloud and on-premises computing resources would be important for cost-contained model development.

Another limitation of the present work also extends to all predictive models: the risk of data drift and changing importance of a predictor over time [42,43]. We acknowledge that diagnostic codes, case definitions, and clinical management may change over time, requiring periodic monitoring for drift in performance. For example, we observed a degradation of calibration when the model was applied to a 2024 cohort, but not model performance based on AUROC. This is perhaps not surprising since factors associated with mortality are unlikely to change drastically over time and AUROC does not rely on calibration. However, the worse calibration for 2024 illustrates that the absolute value of the score can be misleading and should be interpreted cautiously, but its ranking (e.g., discrimination) is less time-varying.

It is possible that the use of variables related to health care resource utilization resulted in some algorithmic bias and therefore different degrees of accuracy for different sociodemographic groups [44]. For example, if select populations are less likely to engage with the healthcare system, are less likely to have a diagnosis rendered, or have the diagnosis rendered at a more advanced stage of illness, then predictors like healthcare utilization frequency may perpetuate such biases [45–47]. We would anticipate underperformance (e.g., higher than expected mortality rates) for such groups. Checklists for artificial intelligence algorithms are available to improve the transparency of artificial intelligence and machine learning models, but additional guidelines are available to help mitigate bias (IJMEDI Checklist in S1 File) [48].

Importantly, we expect these findings to be generalizable to other jurisdictions because comorbidity indices have been demonstrated to be similarly prognostic internationally [49,50]. We also expect the model to generalize to outcomes that are correlated with 1-year mortality. However, we expect performance to deteriorate as look-forward windows increase (e.g., 5-year mortality). Our results are also likely generalizable to disease-specific cohorts, but we expect worse performance because disease-specific or event-specific indicators are likely more important as they become more prevalent. A similar reasoning can be made to explain the worse performance for outcomes like hospital readmission, potentially requiring different models to be trained for specific outcomes [34,51].

Another limitation is the worse performance of the model for predicted probabilities of dying ≥70%, although this represents a small absolute number of people (n = 543; 0.01% of the adult population). The performance of the model is expected to be better if more granular clinical indicators were available. For example, we demonstrated that the specific cancer type is an important prognostic factor that improved model performance (accuracy and calibration), but further breakdown by cancer stage, cancer subtype, and even method of detection (screening versus symptomatic) is expected

to yield further improvements. Similar arguments can be made for other diseases or measures of general health beyond ADL, but such data are unavailable or incomplete in our databases.

Another limitation is inherent to errors in administrative coding for diagnoses. Inaccuracies for coding renal disease have been noted, and algorithms for more accurate case definitions have been published for diabetes [52], congestive heart failure [53], ischemic heart disease [54], stroke [55], and upper gastrointestinal diseases [56]. However, none of these algorithms are perfect and a search for optimal definitions of comorbidities becomes onerous and perhaps unnecessary for the present scope, which aims to model mortality rather than to measure the association of each comorbidity with mortality.

## Conclusion

Tree-based ensemble methods yielded the most accurate predictive model for 1-year mortality using individual comorbidities and additional measures of health and healthcare utilization for the general population. We provide algorithms that other investigators can use to determine a persons' baseline risk of mortality that can be used for direct study or for risk adjustment.

## Supporting information

**S1 File. STROBE checklist for observational studies in epidemiology.** International Journal of Medical Informatics (IJMEDI) checklist checklist for assessment of medical artificial intelligence.
(PDF)

**S2 File. Includes supplementary tables for administrative codes, feature definitions, Python code for training/testing the CatBoost model 1H, and supporting results.**
(DOCX)

**S3 File. CatBoost Model 1H.**
(CBM)

**S4 File. CatBoost Model 2E.**
(CBM)

## Acknowledgments

Parts of this material are based on data and information compiled and provided by CIHI. However, the analyses, conclusions, opinions, and statements expressed herein are those of the author, and not necessarily those of CIHI. Parts of this publication are based on data provided by ICES. However, the views expressed in this publication are those of the researcher and do not necessarily represent those of ICES. This report was produced with the support of the Ontario Ministry of Health. However, the views expressed herein are those of the author, and not necessarily those of the Ontario Ministry of Health or the Government of Ontario.

## Author contributions

**Conceptualization:** Steven Habbous, Erik Hellsten.

**Formal analysis:** Steven Habbous.

**Investigation:** Steven Habbous, Peter C. Austin, Shabnam Balamchi, Erik Hellsten.

**Methodology:** Steven Habbous, Peter C. Austin, Shabnam Balamchi, Davood Astaraky, Roozbeh Yousefi, Munaza Chaudhry, Erik Hellsten.

**Supervision:** Erik Hellsten.

**Validation:** Steven Habbous, Peter C. Austin, Shabnam Balamchi, Davood Astaraky, Roozbeh Yousefi, Munaza Chaudhry, Erik Hellsten.

**Visualization:** Steven Habbous.

**Writing – original draft:** Steven Habbous.

**Writing – review & editing:** Peter C. Austin, Shabnam Balamchi, Davood Astaraky, Roozbeh Yousefi, Munaza Chaudhry, Erik Hellsten.

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
