## [Decision Letter · Decision Letter 0]

10 Feb 2026

Dear Dr. Habbous,

We look forward to receiving your revised manuscript.

Kind regards,

Suyan Tian

Academic Editor

PLOS One

Journal Requirements:

Reviewers' comments:

Reviewer's Responses to Questions

**Comments to the Author**

1. Is the manuscript technically sound, and do the data support the conclusions?

Reviewer #1: Yes

2. Has the statistical analysis been performed appropriately and rigorously?

Reviewer #1: Yes

3. Have the authors made all data underlying the findings in their manuscript fully available?

Reviewer #1: Yes

4. Is the manuscript presented in an intelligible fashion and written in standard English?

Reviewer #1: Yes

Reviewer #1: 1. Hyperparameter Tuning: The manuscript notes only manual tuning due to resource constraints. For a study with millions of records, consider at least grid or randomized search for the best-performing model (CatBoost). Was early stopping or validation monitored robustly?

2. Handling of Missing Data: While categorical variables were said to be non-missing, please clarify the handling of missingness in continuous variables (e.g., ADL, lab values). Was imputation used or were records dropped?

3. External Validation / Generalizability: All results are Ontario-specific. Are there plans or suggestions for external testing (e.g., other provinces or countries)? Is the model publicly available for replication/applications elsewhere, in line with open science principles?

4. Interpretability vs. Accuracy: EBM models are included, but performance is lower than CatBoost. In settings where clinical application is key, how do you envision balancing performance with “glass-box” interpretability? Can high-importance features be mapped to actionable recommendations?

5. Calibration at High Predicted Risk: The authors rightly note some miscalibration among highest-risk individuals (70-100%). Are there possible remedies—e.g., post-hoc recalibration methods, binning, or adjustment using more granular clinical predictors?

6. Socio-demographic Predictors: The feature set is impressive but does not mention the inclusion of socio-economic or geographic variables. Would their addition improve model performance or public health utility?

7. Feature Selection: Given redundancy among health utilization indicators, was collinearity assessed? Are certain features redundant, and could a more parsimonious model perform similarly?

8. Limitations Section: While strengths are well-publicized, the discussion of limitations (data access, coding accuracy, generalizability, computational efficiency, bias sources) could be expanded.

.

Reviewer #1: No

---

## [Author Response · Author response to Decision Letter 1]

24 Feb 2026

February 23, 2026

Re: PONE-D-25-39944.R1 “Using tree-based ensemble methods to produce a population-based mortality risk score in Ontario, Canada”

Dear Dr Tian,

On behalf of all authors, we thank you and the reviewers for the review and suggestions to improve the manuscript. We provide point-by-point responses below. Any other changes were to meet PLOS ONE’s stylistic requirements. We believe the manuscript has been strengthened following these revisions.

Sincerely,

Steven Habbous, PhD

Epidemiologist

Email: shabbous@uwo.ca; steven.habbous@ontariohealth.ca

Reviewer Comments

Reviewer #1:

Comment #1: Hyperparameter Tuning: The manuscript notes only manual tuning due to resource constraints. For a study with millions of records, consider at least grid or randomized search for the best-performing model (CatBoost). Was early stopping or validation monitored robustly?

Response: Thank you for the questions about hyperparameter tuning. Early stopping was explored in model 2C. In the present revision, we expand this to increase the number of iterations to 3000 while maintaining depth=6 and a learning rate of 0.05, with early stopping at 100 iterations and using the AUROC instead of the log-loss as the evaluation criterion to determine early stopping. From this analysis, the optimal number of iterations was 2992, which produced an AUROC of 0.931, which is similar to 1000 iterations (but took 3x as long to run). In addition, all sensitivity analyses seemed to converge around similar estimates (AUROC 0.93, PR-AUC 0.29, Brier Score 0.003, and ICI 0.0003), suggesting the model has achieved saturation and any improvements are likely to be subtle. Regarding the remaining hyperparameter (minimum number of observations in a leaf), we do not feel that reducing this is warranted (may lead to overfitting and proportions with a denominator <10 seem unstable). We have increased this using a manual grid using 20 and 50, but this did not influence the results (added as a footnote to Table 2).

Comment #2: Handling of Missing Data: While categorical variables were said to be non-missing, please clarify the handling of missingness in continuous variables (e.g., ADL, lab values). Was imputation used or were records dropped?

Response: Thank you for this question. Categorical variables were non-missing because there are only codes available to indicate presence of a state (e.g., heart failure) rather than its absence. A similar rationale was used for continuous variables (e.g., counts of healthcare service visits). People with no lab values (e.g., serum creatinine) were assumed to not have chronic kidney disease and were assigned as such. We believe this is justifiable since lab tests are the main mechanism for diagnosis. In contrast, persons only completed a survey having ADL information available if they received the InterRAI surveys (e.g., home care participants or residents of long-term care facilities). As a results of this selection, those missing this survey are not missing at random, or even close to random, as we suspect that survey recipients are more frail than the general population. In light of this violation of the assumptions for multiple imputation, we opted instead to assign them their own category. We have added some of these details to the methods section.

Comment #3: External Validation / Generalizability: All results are Ontario-specific. Are there plans or suggestions for external testing (e.g., other provinces or countries)? Is the model publicly available for replication/applications elsewhere, in line with open science principles?

Response: Although our results are Ontario-specific, we do expect generalizability to jurisdictions having similar healthcare systems as Ontario (e.g., universal health systems) because the risk of death is more a function of biology (e.g., age, comorbidities, general health) than it is of geography. Comorbidities indexes have been in use for decades in multiple countries, yet only small changes have been suggested that only marginally influence their accuracy (e.g., HIV/AIDS).

To partly address this, we created a cohort of Ontarians alive as of January 1, 2024 and applied Model 2E to this new validation cohort. An important observation is that the model was equally performant based on AUROC 0.932 (versus 0.932), but slightly worse on PR-AUC 0.254 (versus 0.290), with some miscalibration (ICI 0.0008 versus 0.0004). The effect on calibration is better shown on the calibration plot below. We have added this to the results section with this new figure (Figure 4).

We also add the following to limitations subsection of the discussion:

“we observed a degradation of calibration when the model was applied to a 2024 cohort, but not model performance based on AUROC. This is perhaps not surprising since factors associated with mortality are unlikely to change drastically over time and AUROC does not rely on calibration. However, the worse calibration for 2024 should caution users of such risk scores that the absolute value of the score can be misleading, but its ranking is less time-varying.”

We append the model as a CatBoost extension (.cbm) as well as a more reader-friendly JSON file and sample code in the appendix. We introduce a new appendix table 3 to provide more details on the features used.

Comment #4: Interpretability vs. Accuracy: EBM models are included, but performance is lower than CatBoost. In settings where clinical application is key, how do you envision balancing performance with “glass-box” interpretability? Can high-importance features be mapped to actionable recommendations?

Response: Prediction models only need to be interpretable if the use-case requires it. Rather than replace the more accurate model, the interpretability component is included to lend credibility and support to the model’s construction and outputs. A predictive model that does not need to be interpretable should not be degraded for the sake of explainability. However, the onus is on the end-user to make this distinction. For example, one study used a mortality risk score computed at the time of admission (https://pubmed.ncbi.nlm.nih.gov/31253736/). Patients with a score exceeding some threshold (0.1) were referred for palliative care consultation, but the actual risk score was not provided because the decision to provide palliative care should be left to the discretion of the clinical staff, not the model. In that example, the score itself was not important, nor were the individual components that were the main drivers behind this score. This is a nice example of the distinction between data-driven and data-informed decision-making. We added the following to the discussion:

“Yet another approach would be to take a data-informed rather than a data-driven approach. For example, one study computed a mortality risk score at the time of admission, and patients with a score exceeding some threshold were referred for palliative care consultation (PMID 31253736). However, the actual risk score was not provided because the decision to provide palliative care should be left to the discretion of the clinical staff, not the model. In this example, neither the score itself was important (only the threshold chosen), nor were the individual components that were the main drivers behind risk score.”

Comment #5: Calibration at High Predicted Risk: The authors rightly note some miscalibration among highest-risk individuals (70-100%). Are there possible remedies—e.g., post-hoc recalibration methods, binning, or adjustment using more granular clinical predictors?

Response: There are indeed possible remedies, but we hesitate to employ them because of the small sample size of the people affected (n=543). We have added the following to the limitations section of the discussion in addition to a sensitivity analysis (model 2G) that includes cancer stage:

“Another limitation is the worse performance of the model for predicted probabilities of dying ≥70%, although this represents a small absolute number of people (n=543; 0.01% of the adult population). The performance of the model is expected to be better if more granular clinical indicators were available. For example, we demonstrated that the specific cancer type is an important prognostic factor that improved the model performance (accuracy and calibration), but further breakdown by cancer stage, cancer subtype, and even method of detection (screening versus symptomatic) is expected to yield improvements. Similar arguments can be made for other diseases or measures of general health beyond ADL, but such data are unavailable or incomplete in our databases.”

Comment #6: Socio-demographic Predictors: The feature set is impressive but does not mention the inclusion of socio-economic or geographic variables. Would their addition improve model performance or public health utility?

Response: This is a great question. Including sex in the model is defensible because there are sex-specific risks (e.g., delivery; prostate cancer). Moreover, age is a non-modifiable and critical risk factor for mortality that should be included. For population-based studies of risk like ours, it may not be appropriate to include measures of race/ethnicity, socioeconomic position, or geography. Instead, further reporting model performance across these subpopulations using the predictive model constructed for the entire population can reveal differences that warrant further scrutiny: it is more important to estimate the effect of these features than to adjust for them. Their inclusion in the model would improve model performance but at the disservice of public health utility.

Comment #7: Feature Selection: Given redundancy among health utilization indicators, was collinearity assessed? Are certain features redundant, and could a more parsimonious model perform similarly?

Response: While there may be some redundancy and collinearity between features, this is only a limited concern because this will not impact the accuracy of the model. Instead, it may influence the models’ interpretability. For example, if A and B were highly colinear, sometimes a tree will split on A and other times on B instead of A, but will rarely split on both. The end result is that the importance of A and B will be lower than either of their true importance, but since the split is still being made, the models’ prediction would be unchanged. For demonstration, I created a new variable Age2 = Age plus random variation (drawn from a standard normal distribution). Adding Age2 to the model (very highly linearly correlated with age) produced fit statistics equivalent to the original model (AUROC 0.932; PR-AUC 0.287; Brier Score 0.0082; ICI 0.0003). However, regarding feature importance, the original “Age” importance was reduced by approximately half, with “Age2” stealing that half. This figure is shown below (the original from Figure 3A is on the right for comparison):

A more parsimonious model would have the effect of reducing performance while potentially improving explainability. However, since we weigh performance in this case to be more important than explainability, a model with fewer predictors is of limited utility. We do, however, provide a few options. The first is a model that treats each comorbidity as a dichotomy (absent/present) rather than distinguishing the presence based on the source of data (model 2D versus 2A). This would greatly simplify data preparation with little impact on model performance. The second is a model that only includes a minimal set of datasets available to most researchers: hospital data, physician billing data, and cancer registry data. We removed ADL from the model and added this sensitivity analysis to Table 2 as Model 2H.

Comment #8: Limitations Section: While strengths are well-publicized, the discussion of limitations (data access, coding accuracy, generalizability, computational efficiency, bias sources) could be expanded.

Response: We have expanded the limitations section to discuss some of these issues mentioned above. In addition, we also discuss the accuracy of diagnostic codes around case definitions for various comorbidities:

“Other limitations is due to the inherent errors associated with administrative data to code diagnoses. Inaccuracies for coding renal disease have been noted, and algorithms for more accurate case definitions have been published for diabetes (Lipscombe et al., 2018), congestive heart failure (Schultz et al., 2013), ischemic heart disease (Tu et al., 2010), stroke (Hall et al., 2016), and upper gastrointestinal diseases (Lopushinsky et al., 2007). However, none of these algorithms are perfect and a search for optimal definitions of comorbidities becomes onerous and perhaps unnecessary for the present scope, which aims to model mortality rather than to measure the association of each comorbidity with mortality.”

We also added a paragraph about computational resources:

“One limitation is the computational resources available. While advancing our ability and capacity to use machine learning methods, our organization is still immature on this front [6]. We adhere to a 'Cloud-First' strategy, but the reliance on a specific-vendor model imposes structural and financial constraints that are ill-suited for the iterative nature of machine learning research. On our Azure Virtual Desktop we have access to eight CPU cores, 137 Gb of RAM, and no GPUs. We were unable to conduct full cross-validation assessments or hyperparameter tuning through grid or randomized search, and it remains possible that a better model fits these data. Despite this, the convergence of different models across all measures of model performance suggest that any further improvements in the model are likely to be small. For the present study this was not a barrier, but for work requiring more sophisticated models, a hybrid approach of cloud and on-premises computing resources would be important for cost-contained model development.”

---

## [Decision Letter · Decision Letter 1]

31 Mar 2026

Using tree-based ensemble methods to produce a population-based mortality risk score in Ontario, Canada

PONE-D-25-39944R1

Dear Dr. Habbous,

We’re pleased to inform you that your manuscript has been judged scientifically suitable for publication and will be formally accepted for publication once it meets all outstanding technical requirements.

Kind regards,

Suyan Tian

Academic Editor

PLOS One

Additional Editor Comments (optional):

Reviewers' comments:

Reviewer's Responses to Questions

**Comments to the Author**

Reviewer #1: All comments have been addressed

2. Is the manuscript technically sound, and do the data support the conclusions?

Reviewer #1: Yes

3. Has the statistical analysis been performed appropriately and rigorously?

Reviewer #1: Yes

4. Have the authors made all data underlying the findings in their manuscript fully available?

Reviewer #1: No

5. Is the manuscript presented in an intelligible fashion and written in standard English?

Reviewer #1: Yes

Reviewer #1: Gemini said

The authors have successfully addressed the reviewers' concerns by providing a more transparent view of the CatBoost model's implementation and its temporal stability. The addition of the 2024 validation cohort is a significant improvement that clarifies the model's predictive limits and calibration drift over time. Furthermore, the detailed justification for the feature selection process and the data availability statement, which correctly balances PHIPA legal restrictions with the provision of the model's JSON code, ensure the study meets the journal's standards for reproducibility and ethical rigor. The manuscript is now suitable for publication.

.

Reviewer #1: No

---

## [Editor Report · Acceptance letter]

PONE-D-25-39944R1

PLOS One

Dear Dr. Habbous,

I'm pleased to inform you that your manuscript has been deemed suitable for publication in PLOS One. Congratulations! Your manuscript is now being handed over to our production team.

Kind regards,

on behalf of

Dr. Suyan Tian

Academic Editor

PLOS One